# Association between Psychopathological Symptoms and Aggression and Selected Biochemical Parameters in Adolescents with Behavioural and Emotional Disturbances

**DOI:** 10.3390/ijms24087097

**Published:** 2023-04-12

**Authors:** Małgorzata Śmiarowska, Małgorzata Pawlicka, Agnieszka Boroń, Anna Grzywacz, Krzysztof Safranow, Dariusz Chlubek, Violetta Dziedziejko

**Affiliations:** 1Department of Pharmacokinetics and Therapeutic Drug Monitoring, Pomeranian Medical University, 70-111 Szczecin, Poland; malgorzata.smiarowska@wp.pl; 2Department of Child and Adolescent Psychiatry, Independent Public Specialist Health Care Center “ZDROJE”, 70-780 Szczecin, Poland; 3Department of Clinical and Molecular Biochemistry, Pomeranian Medical University, 70-111 Szczecin, Poland; 4Independent Laboratory of Health Promotion, Pomeranian Medical University in Szczecin, 70-111 Szczecin, Poland; grzywacz.anna.m@gmail.com; 5Department of Biochemistry and Medical Chemistry, Pomeranian Medical University, 70-111 Szczecin, Poland; chrissaf@mp.pl (K.S.);

**Keywords:** emotional disturbances, BDNF, cortisol, quality of life, adolescent

## Abstract

Behavioural and emotional disturbances (F92.8) are the most recognized disorders in a developmental psychiatry. As the problem is still alarmingly increasing, the searches for their etiopathogenesis and more effective preventing and therapy methods are required. The aim of the study was to assess the association between the quality of life, some psychopathological features, concentrations of selected immunoprotective (brain-derived neurotrophin, BDNF), and endocrine (cortisol, F) factors while adolescent disturbances. The study was performed in 123 inpatients of a psychiatric ward with F92.8 diagnosis, aged 13–18 years. The complete patients’ interview, physical examination, and routine laboratory tests, including serum F and BDNF tests, were performed. All patients completed standardized questionnaires to estimate: the severity of psychopathological symptoms (SCL-90), the level of aggression (Buss–Perry). The changes in the plasma BDNF and F concentrations were shown in patients raised in foster homes and institutions. The significantly lower BDNF was observed in youth from foster and suicide-experienced families. The more severe psychopathological symptoms, especially aggression and hostility, were found in these ones, who abused alcohol, attempted suicide, had lower self-esteem and cognitive processes, and were lacking safety in dysfunctional families.

## 1. Introduction

The period of adolescence and early adulthood is a special stage in human life, because it is connected with changes and developmental crises that everyone must overcome on the way to adulthood. It is filled with new needs, which are often contradictory, to define oneself and one’s place in the peer group, family, or society. Characteristic physiological, emotional, and psychological changes, which take place in the adolescent’s body, determine the quality of their future life. At the cellular level, they are firmly established in the regulation of cytotoxic processes linked with neuroprotective processes. As a result, by the coincidence of unfavourable trigger mechanisms [1], clinical manifestations of diseases and mental disorders may occur, including depressive disorders [2,3,4], school difficulties [5], contact with psychoactive substances [6,7,8], self-destructive behaviours, i.e., self-harm and suicide attempts [9,10], as well as dissociable behaviours [11].

According to the World Health Organization (WHO), 14% of children aged 10–19 years experience mental health disorders, although they remain largely undiagnosed and untreated. Anxiety disorders, which may include panic or excessive worry, are more common amongst older than younger adolescents. According to the WHO data from 2021, 3.6% of adolescents aged 10–14 years and 4.6% of adolescents aged 15–19 years experience anxiety disorders. Depression occurs in 1.1% of adolescents aged 10–14 years and 2.8% of adolescents aged 15–19 years. Depression in adolescents is a major risk factor for suicide, which is the fourth most common cause of death for people aged 15–19 years [12].

The fast pace of life, parents’ employment, including their work-related trips, or being focused on success and self-development, single parenthood, illness and/or either parent’s or both parents’ addiction, family conflicts, and, as a result, insufficient or absent support from the loved ones deprive the minor of their needs and contribute to a decrease in their mental resilience, disordered psychological development and manifestation of mental dysfunctions that require specialist interventions. The family environment in childhood and adolescence is an important link for understanding the mental and physical health of the patient throughout their lifetime. Not only does the period of adolescence, due to its critical phase of transition from childhood to adulthood, create favourable conditions for the emergence of typical adaptive mental problems, but it may also indicate established dysfunctions and syndromes which are treated only in adulthood—when the personal and social costs of the individual are high [13].

Bearing in mind that half of all mental illnesses develop before the age of 14 years, efforts to prevent the onset of mental illness and promote mental health during adolescence are of particular importance [14,15].

The aim of the cross-sectional study was to assess an association between the quality of life, severity of psychopathological symptoms, including, in particular, aggression, and concentrations of selected immunoprotective factors (BDNF), as well as endocrine factors (cortisol) in adolescents with behavioural and emotional disturbances.

## 2. Results

### 2.1. Characteristics of the Study Group

Appendix A presents the sociodemographic characteristics of the patients. Appendix A presents the general clinical characteristics, and Appendix A presents the characteristics of the patients based on the self-description questionnaires used to assess the severity of psychopathological symptoms.

The study group of adolescents obtained high scores on all the subscales of the SCL-90 questionnaire used to measure the severity of psychopathological symptoms, including somatization, compulsiveness, interpersonal hypersensitivity, depression, anxiety, hostility, phobias, and paranoid thinking. The mean score for the study group remained within the average only on the subscale used to assess the severity of symptoms related to psychoticism (Appendix A). Based on the standards adopted for the Polish adolescents, the adolescents under study also achieved high scores on the scale of hostility in the Buss–Perry questionnaire as well as in the assessment of aggressiveness. Higher scores were obtained on the scales of anger and physical aggression. Verbal aggression, however, was at the average level in relation to the entire Polish population (Appendix A).

### 2.2. Comparative Analysis of Biochemical Parameters and Psychological Characteristics of Adolescents, Depending on Their Gender, Age, and Suicide Attempts

In the assessment of biochemical parameters, statistically significant higher concentrations of potassium (4.41 ± 0.34 mmol/L and 4.30 ± 0.33 mmol/L; *p* = 0.045), creatinine (0.72 ± 0.14 mg/dL and 0.61 ± 0.10 mg/dL; *p* < 0.00001), as well as alanine and aspartate aminotransferases (32.6 ± 74.6 U/L and 14.6 ± 7.9 U/L, *p* = 0.000085; 24.1 ± 10.1 U/L and 18.9 ± 5.6 U/L, *p* = 0.000021) in serum were found in boys compared to girls.

In psychological studies using the SCL-90 questionnaire, girls obtained statistically significantly higher results than boys on the scale of somatization, hypersensitivity, and depression (Table 1).

In the group of older adolescents (>15 years of age), a statistically significantly higher serum creatinine concentration (*p* = 0.024) and significantly lower thyrotropin concentration (*p* = 0.020) were found compared to the group of younger subjects (<15 years of age).

A statistically significantly higher degree of somatization, compulsion, and anxiety was found in the group of older adolescents assessed using the SCL-90 questionnaire (Table 1).

The scores obtained using the Buss–Perry questionnaire show no statistically significant differences between the aggressiveness of younger and older adolescents (Table 1).

SCL-90 Questionnaire: SOM_SCL—somatization, OC_SCL—obsessive–compulsive, Sens_SCL—interpersonal sensibility, DEP_SCL—depression, ANX_SCL—anxiety, ANG_SCL—anger–hostility, PHO_SCL—phobic-anxiety, PAR_SCL—paranoid ideation, PSY_SCL—psychoticism. Buss–Perry Questionnaire: A_BP—anger, PA_BP—physical aggression, H_BP—hostility, VA_BP—verbal aggression, SUM_BP—sum.

The adolescents under study with suicide attempts in their history obtained statistically significantly higher scores, which indicated a greater severity of somatization, depression, anxiety, paranoid thinking, psychoticism, as well as hostility compared to the individuals with no history of suicide attempts (Table 2).

No statistically significant differences were found in the general clinical and biochemical parameters between adolescents who attempted suicide and those who did not.

SCL-90 Questionnaire: SOM_SCL—somatization, OC_SCL—obsessive–compulsive, Sens_SCL—interpersonal sensibility, DEP_SCL—depression, ANX_SCL—anxiety, ANG_SCL—anger–hostility, PHO_SCL—phobic-anxiety, PAR_SCL—paranoid ideation, PSY_SCL—psychoticism. Buss–Perry Questionnaire: A_BP—anger, PA_BP—physical aggression, H_BP—hostility, VA_BP—verbal aggression, SUM_BP—sum.

### 2.3. Comparative Analyses of General Clinical, Biochemical, and Psychological Characteristics of Patients in Relation to Smoking and Alcohol Consumption as Well as Drug and Designer Drug Use

No statistically significant differences were found in the general clinical, biochemical, or psychological parameters assessed using the SCL-90 and Buss–Perry questionnaires between adolescents admitting to smoking and nonsmokers.

Adolescents who reported alcohol use in their medical history—but failed to meet the addiction criteria—had significantly higher scores on the scale of hostility (1.7 ± 1.0 vs. 1.2 ± 0.9; *p* = 0.015), paranoid thinking (1.6 ± 1.0 vs. 1.3 ± 1.1; *p* = 0.036), and physical aggression (26.5 ± 8.9 vs. 22.3 ± 8.0; *p* = 0.0075), which translated into higher total aggression (90.5 ± 19.1 vs. 82.7 ± 20.8; *p* = 0.044) compared to their peers who declared that they consumed no alcohol.

Adolescents who declared that they took drugs or designer drugs, were statistically significantly older (aged 16.1 ± 1.2 years vs. 15.2 ± 1.3 years; *p* = 0.0019).

In the study using self-description questionnaires, no significant statistical differences in the analysed psychometric parameters were found between individuals who declared that they took drugs and/or designer drugs and individuals who denied taking them.

In the detailed analysis of selected drugs (according to the case history) and designer drugs, the following statistically significant differences were found: serum creatinine levels in patients who took tetrahydrocannabinol (THC) or amphetamines were higher than in patients who took no drugs (THC: 0.70 ± 0.12 mg/dL, vs. 0.62 ± 0.12 mg/dL; *p* = 0.0022; amphetamines: 0.70 ± 0.12 mg/dL vs. 0.63 ± 0.12 mg/dL; *p* = 0.034). Patients who took THC or amphetamines were significantly older compared to patients who took none of these substances (THC: aged 16.1 ± 1.2 years vs. 15.2 ± 1.3 years; *p* = 0.0019; amphetamines: aged 16.1 ± 1.1 years vs. 15.3 ± 1.4 years; *p* = 0.045). Patients who took amphetamines had statistically significantly higher scores for total aggression (Buss–Perry questionnaire) compared to patients who took no amphetamines (94.4 ± 17.6 vs. 84.1 ± 20.6; *p* = 0.045). Patients who used methamphetamines were hospitalized for a statistically significantly longer period of time compared to patients who did not take this substance (32.5 ± 5.7 days vs. 19.5 ± 11.8 days; *p* = 0.022). Serum TSH concentration levels were statistically significantly lower in patients who took mephedrone compared to patients who did not take this substance (1.3 ± 0.6 µIU/mL vs. 2.1 ± 1.0 µIU/mL, *p* = 0.034).

### 2.4. Comparative Analyses of General Clinical, Biochemical, and Psychological Characteristics of Patients, Depending on the Family Situation

Three groups of patients under study were distinguished, depending on their place of residence. The largest group comprised adolescents living with their parents (n = 79) and children in foster homes (n = 20) or institutions (children’s homes, youth sociotherapy centres, youth care centres, boarding houses) (n = 24). Individuals living in institutions had the highest levels of cortisol and BDNF in their blood. A statistical significance was obtained by comparing the subgroup of individuals living in foster homes and institutions (cortisol: 153.6 ± 54.2 μg/L vs. 107.0 ± 49.9 μg/L, *p* = 0.0083; BDNF: 2675 ± 2126 pg/mL vs. 1774 ± 2232 pg/mL, *p* = 0.017).

Significantly lower values of traits, including compulsion (1.0 ± 1.0 vs. 1.5 ± 1.1, *p* = 0.016), depression (1.3 ± 1.1 vs. 1.9 ± 1.0, *p* = 0.019), and phobia (0.5 ± 0.9 vs. 1.1 ± 1.0, *p* = 0.00073) in the SCL-90 questionnaire, were obtained by adolescents raised in institutions compared to adolescents raised in families. On the Buss–Perry scale, adolescents living in institutions showed significantly higher scores for physical aggression compared to children raised by their parents (26.9 ± 7.6 vs. 22.6 ± 8.9, *p* = 0.023).

No statistically significant differences in the general clinical parameters or psychological characteristics assessed using the SCL-90, Buss–Perry questionnaires, were found between adolescents who had a family history of suicide and adolescents with no suicide in the family. However, a significantly lower plasma concentration of BDNF was found in adolescents who had a family history of suicide vs. adolescents from families unburdened with a suicide crisis (BDNF: 496 ± 372 pg/mL vs. 2505 ± 2351 pg/mL, *p* = 0.0014).

Minors who reported that they were unburdened with the problem of alcohol abuse by their parents had statistically significantly higher scores on the subscale of compulsion, interpersonal hypersensitivity, depression, phobia, and psychoticism (assessed using the SCL-90 questionnaire) compared to adolescents who had an alcohol problem in their families. However, no statistically significant differences between the subgroups under study were found for each analysed psychological parameter assessed using the Buss–Perry questionnaire (Table 3).

Children raised in families with no record of mental illness were found to have a statistically significantly lower number of siblings (1.8 ± 1.7 vs. 2.9 ± 2.4, *p* = 0.021) compared to families in which at least one member was diagnosed with mental illness.

No statistically significant differences were found in the analysed biochemical parameters or psychological characteristics assessed using the SCL-90 and Buss–Perry questionnaires between adolescents raised in families without mental illness and adolescents raised in families where this problem existed (Table 3).

SCL-90 Questionnaire: SOM_SCL—somatization, OC_SCL—obsessive–compulsive, Sens_SCL—interpersonal sensibility, DEP_SCL—depression, ANX_SCL—anxiety, ANG_SCL—anger–hostility, PHO_SCL—phobic-anxiety, PAR_SCL—paranoid ideation, PSY_SCL—psychoticism. Buss–Perry Questionnaire: A_BP—anger, PA_BP—physical aggression, H_BP—hostility, VA_BP—verbal aggression, SUM_BP—sum.

### 2.5. Assessment of Correlations between Psychological Traits in Patients under Study

Significant, positive correlations between all the scales with the traits that describe aggressiveness, such as anger, hostility, and verbal aggression, assessed using the Buss–Perry questionnaire were found between psychological dimensions assessed using the SCL-90 questionnaire (Table 4).

Positive correlations were found between the severity of somatization (RS = 0.19), compulsion (RS = 0.22), anxiety (RS = 0.23), and age.

RS—Spearman ‘s rank correlation coefficient, statistically significant correlations are marked in bold, *p* < 0.05.

SCL-90 Questionnaire: SOM_SCL—somatization, OC_SCL—obsessive–compulsive, Sens_SCL—interpersonal sensibility, DEP_SCL—depression, ANX_SCL—anxiety, ANG_SCL—anger–hostility, PHO_SCL—phobic-anxiety, PAR_SCL—paranoid ideation, PSY_SCL—psychoticism. Buss–Perry Questionnaire: A_BP—anger, PA_BP—physical aggression, H_BP—hostility, VA_BP—verbal aggression, SUM_BP—sum.

## 3. Discussion

The observations made by the National Survey of Children’s Health (NSCH) in the US have shown that a dominating psychopathology amongst patients aged 3–17 years included emotional (7.1%—anxiety, 3.2%—depression) and behavioural/conduct (7.4%) disturbances. Their prevalence was dependent on poorer child or parent/caregiver mental/emotional health [16]. The most common reasons for hospitalization in the wards of child and adolescent psychiatry are self-abuse and/or aggression states, anxiety and depressive disorders, acute transient psychotic states, mainly induced by psychoactive substances [3,6,8,10,11]. However, the COVID-19 pandemic restrictions have shown that the number of children and adolescents who were affected by social distancing, school closure, and isolation finally had been negatively impacted in their mental health and well-being mainly concerning anxiety, depression, sleep and appetite disturbances, and social interactions [17].

The dysfunctional family model was found in 87% of the patients from the group of patients with behavioural and emotional disturbances covered by this study. These observations are consistent with the conclusions of other studies that the family context is an indisputable trigger for various types of mental disorders from early childhood to adulthood [18,19,20]. Exposure to early childhood trauma results in disorders that usually manifest during the developmental period [21]. Particularly important components that shape the biological conditions of a child, such as their temperament or search for stimulation, may include mental illness, addiction, divorce, separation, hostility, emotional rejection, neglect, or child maltreatment [22,23]. In the group of adolescents subjected to the current research, 49.6% grew up in families with an alcohol problem, 29.3% in an atmosphere of mental illness, and 5.7% experienced the suicide of a family member. Other dysfunctions of the family system, including marital conflicts, single parenting, or one parent working in another country, were also found.

These strains are documented predictors of the risk of developing mental disorders in children, such as, in particular, worse regulation of emotions and behavioural modulation [22,23] that predispose to abnormal personality development (with borderline or dissociable characteristics) [24,25], affective disorders such as depression, anxiety, and disruptive mood dysregulation disorder (DMDD), as well as predisposition to various addiction and self-destructive behaviours [26]. In the group of children raised in a dysfunctional system and with care and emotional deprivation, cognitive disorders with decreased ability to abstract reasoning, problem-solving skills, and work planning were also found [18,19,27].

In an Iranian general population-based study, which was evaluated according to sociodemographic characteristics in 30,532 children and adolescents (aged 6–18 years), the data have shown that 0.58% of children aged 6–9 years, 0.57% of adolescents aged 10–14 years, and 1.22% of adolescents aged 15–18 years met the criteria of conduct disorder (CD). Amongst the individuals with conduct disorder (mostly found in boys), 83.4% met the criteria for at least one more psychiatric disorder, including oppositional defiant disorder (54.89%), attention-deficit/hyperactivity disorder (32.34%), tobacco use (20.43%), and depressive disorders (18.30%). CD was significantly less prevalent amongst those individuals whose fathers had no history of psychiatric hospitalization [25]. On the other hand, Collishaw, S. et al. [28] have proved that mental health problems in adolescents are common, but not inevitable, even when parental depression is severe and recurrent. In their study on early adolescents, 20% of offspring (aged 9–17 years) had parents with recurrent depression [28].

For the most part, the current study group consisted of adolescents (75.6%) aged 15–18 years. The majority were girls (66.7%) and individuals living in large cities (about 40%) who use recreational drugs and stimulants (56%). Note that 39% of adolescents admitted using psychoactive substances, 68.3% admitted doing self-harm, and 36.6% confirmed suicide attempts. The case history confirmed that the profile of hospitalized children and adolescents reflected their disturbed relationships in families, peer, and school environments that overlapped with adolescence. This is consistent with research studies by other authors who identified environmental conditions as crucial for shaping the brain and building proper mental resilience in the developmental period and the process of individualization—separation [29,30]. Unfortunately, reports by HBSC (Health Behaviour in School-Aged Children) inform that, each year, support for both adolescent boys and girls from the family decreases with age [31]. It directly results in the increasing number of mental problems experienced by children and adolescents and the increasing percentage of suicides [12].

Three groups of patients under study were distinguished, depending on their place of residence. The largest group included adolescents living with their parents. Two comparatively big subgroups of about 20 individuals lived in either foster homes or institutions. Adolescents raised in institutions were more physically aggressive than minors raised in family homes or foster homes. It is believed that increased physical aggressiveness in adolescents raised in institutions is related to the feeling of emotional and existential loneliness that they experience [32], the presence of deficits in emotional control and immaturity of mechanisms connected with self-control [33], as well as the influence of socially dysfunctional environments [34,35]. One may formulate a hypothesis that the upbringing of children in nonfamily environments may promote modulation of externalisation behaviours, such as aggressive discharge of tension and the development of behavioural disturbances.

However, compared to children in institutions, in the group of adolescents raised in biological families—87% of which were dysfunctional—more severe psychopathological symptoms associated with intrapsychic mechanisms (compulsion, depression, phobias) were found. According to psychoanalytic concepts, they are to save the image of the family at the expense of psychosomatic disorders and neurotic mechanisms [36], because connections between the closest members of the family are very strong, and behaviour of one of its members affects the others. That is why, in order to maintain the family system as a whole, many psychopathological symptoms are exacerbated in children at the expense of stopping the process of individualization—separation [36]. In the case of children who are disconnected from the direct influence of dysfunctional parents (placement in a foster home or institution), these influences decrease and the child is part of another system, which may weaken the severity of psychopathological symptoms [37].

The influence of parents’ health on the development of mental disorders in offspring is emphasized in developmental psychology. It was observed that adolescents raised in families with alcohol abuse and/or parent’s mental illness had a statistically significantly higher number of siblings. These observations are consistent with the well-known phenomenon of numerous children in dysfunctional families, especially those with an alcohol problem. The effect that alcohol exerts is related to impaired regulation and control of desires, including sexual desire, as well as disinhibition of violent, aggressive, and interdependent behaviours, co-dependent partnerships, risk-taking, and criminal behaviours (also sexually motivated) [38].

In the current research studies, minors who did not report alcohol abuse by their parents had statistically significantly higher scores on the SCL-90 subscales, indicating a higher severity of psychopathological symptoms, such as compulsion, interpersonal hypersensitivity, depression, phobia, and psychoticism, compared to adolescents who had an alcohol problem in their families. The results show that regardless of the type of dysfunction in the family (alcoholism, parent’s illness), the process of solving developmental problems in a minor is disturbed. However, because of struggling with numerous stressful events, the child may build special mental distance from the situation in which they find themselves, which is based on their own resources and at least to increase their adaptability (the phenomenon of resilience) [39].

As far as is known, more than 80% of children who attempt suicide are not identified by paediatricians in a routine visit months before their suicide attempt [40]. To date, no other specific instruments have been found to determine the risk of suicide in situations of depression and suicide threats. To identify such ones seems to be especially important as it has been documented that the prevalence of depression and mental health disorders has been increasing, and depression is still a common mental health disorder in children and adolescents, affecting around 3% of younger children and about 8% of adolescents [40].

Psychological mechanisms and the dynamics of mental processes regulations are supposed to be partially reflected in biological parameters. Therefore, BDNF (brain-derived neurotrophic factor) and cortisol were chosen to assess some of the psychoendocrine processes in this study.

BDNF plays a key role in the survival, differentiation, morphogenesis, and synaptic plasticity of neurons in ontogeny. It affects the central and peripheral nervous system [41]. In addition, it is related to the development of serotonergic, dopaminergic, noradrenergic, and cholinergic neurons [42]. The largest amount of BDNF was found in the limbic system (mainly the hippocampus), the cerebral cortex (prefrontal cortex), and the cerebellum. These structures are known to be responsible for the regulation of emotions and participation in memory processes [43]. BDNF levels may be affected by many factors, including age, gender, as well as genetic, environmental, and hormonal factors.

Genetic variations can play an important biological role in the neuroplasticity process. SNP rs6265 in the *BDNF* gene represents one of the most studied polymorphisms for its role in the regulation of neuronal differentiation and plasticity. SNP rs6265 is the substitution of guanine by adenosine at nucleotide 196, resulting in a change from valine to methionine at codon 66 (Val66Met). This modification does not promote alteration in the neuronal structure, but it is related to a decrease in neuronal BDNF concentration, changes in intracellular traffic, and secretion of mature BDNF and, as a consequence, a final reduction of BDNF levels by approximately 50%. Reduction in volume of the hippocampus has been associated with the Val66Met polymorphism, even in healthy subjects [44].

A genetic variant rs6265 impacting on emotion processing is known to increase the risk for depression. Gatt et al. observed that individuals carrying the *BDNF* Met allele who had been exposed to early life stress presented reduced gray matter in the hippocampus [45]. In this study, the size of the hippocampus correlated with reduced lateral prefrontal cortex volume and higher depression scores. Association of the common *BDNF* gene rs6265 functional polymorphism with depressive symptoms has been reported [46]. We are also currently conducting research on the impact of rs6265 and other *BDNF* gene polymorphisms on behavioral and emotional disorders in adolescents, the results of which will be published soon.

The role of BDNF in the aetiology of mental and neurological diseases is also reported as significant [47]. In the literature, there are no data showing BDNF levels in adolescents depending on the form of care they are provided with as well as environmental conditions. The current study is the first to determine the concentration of BDNF in children with behavioural and emotional disturbances in foster homes and institutions compared to adolescents burdened with living in difficult family conditions (dysfunctional biological family). The largest study group included adolescents living with their parents. However, the highest levels of cortisol and BDNF in the blood were found in people living in institutions, and they were statistically significantly higher compared to patients raised in foster homes. The above observations are consistent with the results obtained by Fischer et al., who indicated a lower concentration of cortisol in children raised in foster homes [48] and burdened with a serious family conflict [49]. However, in children raised in foster homes on a regular basis, daily fluctuations in cortisol were lower compared to children who lived in foster homes only in emergency situations [48]. Cortisol secretion is an indicator of the activity of the hypothalamic–pituitary–adrenal axis (HPA) with significant circadian variability depending functionally on disorders related to stress, insomnia, depression, and anxiety. Adolescence is connected with adaptation to new situations that require activation of the HPA system, which is indirectly expressed by the release of cortisol.

Based on our analysis, one may formulate a hypothesis that the foster home model adopted on a regular/permanent basis, not temporarily, seems acceptable, stable, and safe, as it generates lower stress for the developing brain. This is confirmed by lower activity of its neuroplasticity that is indirectly assessed with BDNF available in the peripheral study. In the currently conducted observations, no significant difference was found in the concentrations of BDNF and cortisol in groups of adolescents raised in foster homes and institutions compared to the group raised by biological parents. This is a surprising result that requires detailed observation, as it suggests that, in terms of safety and protection from stress, the environments providing care outside the family are similar to the biological family. On the other hand, one may reasonably assume that biological families of patients subjected to this observation—who had emotional disturbances that required psychiatric hospitalization due to the severity of disadaptive symptoms—were similarly “damaging”, threatening or deficient. As a result, the level of stress and tension generated in the biological families of the children under study was like stress and tension in other institutional forms of foster care.

Significantly lower plasma BDNF concentrations were found in the group of adolescents who experienced suicide in their family. Research into genetic determinants of suicide has a very long history, although there are few research studies into the role of BDNF in suicide. Based on the research results, a hypothesis was formulated that BDNF might be recognized as a sign of a suicide crisis [50]. Darlington et al. conducted an analysis of family-related risk factors for suicide and changes in the BDNF concentration amongst children of suicides. They found that the reduced BDNF concentration may therefore be observed not only in people who commit suicide, but also in their offspring [51]. The current studies confirm these observations, and, although the number of patients with a family history of suicide is relatively small (n = 7), detailed analysis performed in a larger group is required.

There are two main limitations of our study: a relatively low number of subjects and lack of a control group of adolescents without behavioural and emotional disturbances. Further studies are needed to confirm associations found in our study group and to relate the measured parameters to a properly matched control group of healthy adolescents.

Although, no significant association between the severity of psychopathological symptoms, including the level of aggression, and blood levels of cortisol and BDNF was found in adolescents (aged 13–18 years) with diagnosed behavioural and emotional disturbances, the results obtained in the study reveal a very important practical dimension—that BDNF in serum may be helpful in verifying children and adolescents who are a suicide risk.

## 4. Materials and Methods

### 4.1. Study Design and Participants

The study covered minors aged 12–18 years from the Department of Child and Youth Psychiatry in Szczecin, after they had provided their own written consent (aged 15 years or older), as well as the consent granted by their legal guardian. The study group consisted of 123 people, including 82 girls (67%) and 41 boys (33%). The mean age of the patients was 15.4 ± 1.3 years; there was no difference in age between girls and boys (*p* = 0.92 Mann–Whitney test). The study was approved by the Bioethics Committee of the Pomeranian Medical University in Szczecin (No. KB-0012/29/15). The patients had a diagnosis of behavioural and emotional disturbances established by a psychiatrist in accordance with the ICD-10 [52]. Patients who participated in the study were treated in the Department of Child and Adolescent Psychiatry. Recruited adolescents were analyzed for psycho/biochemical findings without the influence of concomitant alcohol consumption. Directly before taking blood from the patient and completing the questionnaires, alcohol was not measured; it was not justified, because the patients included in the study were patients from a closed ward of the Department of Child and Adolescent Psychiatry, sober, without access to alcohol and other stimulants. Earlier, a psychiatrist examining a patient in the emergency room refers only sober patients to treatment in a closed ward. In case of a suspicion of insobriety, the measurement is performed by a doctor using a breathalyser and the patient is referred to another ward, a toxicology one.

The exclusion criteria included a confirmed diagnosis of schizophrenia or affective disorders, addiction to psychoactive substances, mild or severe mental retardation, epilepsy, general developmental disorders, cancer, as well as metabolic and autoimmune diseases.

The patients’ case history included their personal profile, medical history, family and social history, as well as physical examination.

### 4.2. Blood Samples and Biochemical Measurements

The tests were performed on the first day after admission. During the study, the patients took no psychotropic drugs. Blood was collected after the night rest, between 6:15 and 7:00 a.m., from the ulnar vein (4 mL for the EDTA anticoagulant tube, and 2.8 mL for the clot activator tube). The samples were centrifuged for 15 min at 4 °C, 1000× *g*. In order to obtain low-platelet plasma, 1 mL of plasma was subjected to centrifugation for 10 min at 4 °C, 10,000× *g*. The BDNF concentration was determined for the plasma. The resulting material (plasma, low-platelet plasma, serum) was protected for analysis at −80 °C.

Analyses of human free BDNF concentrations in plasma were performed using commercially available ELISA tests (Quantikine, R&D Systems Europe, Abingdon, UK, catalogue No. DBD00).

Additionally, routine laboratory tests were performed, including morphology, concentration of sodium, potassium, glucose, creatinine, c-reactive protein test (CRP), thyrotropic hormone (TSH), cortisol, as well as alanine and aspartate aminotransferase activity (ALT, AST).

The concentrations of biochemical parameters were measured in serum, and the concentrations of BDNF in low-platelet plasma.

### 4.3. Questionnaire Analysis

On the first day of hospitalization, the study group completed self-description questionnaires to assess the severity of psychopathological symptoms, including aggression, standardized for the group of individuals aged 12–18 years. The popular international SCL-90 test (Derogatis, Lipman, Covi) in the electronic version, “www.psychologia.net” (accessed on 15 June 2020), adapted by K. Jankowski [53], was used in the study, as well as the Polish version of the Buss–Perry Aggression Questionnaire, which is authorized for research purposes in the electronic and paper version for the general population without the consent from the Amity Institute [54]. The questionnaires were completed in privacy, independently, in the discreet presence of the researcher.

### 4.4. Statistical Analysis

For statistical calculations, nonparametric tests were used, since in most cases the measurable parameter distributions differed significantly from the normal distribution (*p* < 0.05, Shapiro–Wilk test). The parameters were compared between the subgroups of patients using the Mann–Whitney U test. Correlations between measurable parameters were analysed using the Spearman’s rank correlation coefficient. Statistical calculations were performed using Statistica 12. The statistical significance threshold was *p* < 0.05.

## 5. Conclusions

Healthy parenting increases the ability to regulate emotions and promotes a healthy, psychosocial development of children and adolescents. Accurate diagnosis of a disorder, in which the environmental context is taken into account, may protect the minor from the negative effects of early traumatization, which is usually manifested in the conditions of adolescent stress. Parenting and substitute systems, especially when dysfunctional, are an important nonpharmacological area for intervention and prevention of developmental disorders.

An association between the severity of psychopathological symptoms and gender and age of the patients was found in the study. Older adolescents showed greater severity of psychopathological symptoms in the dimension of somatization, compulsion, and anxiety. Girls showed more severe symptoms of somatization, interpersonal hypersensitivity, and depression than boys. These differences may indicate the influence of many biological, psychological, and social variables on the adolescent.

In adolescents with a history of suicide attempt, intensification of somatization, depression, anxiety, paranoid thinking, and psychoticism, as well as their significantly higher intensity of hostility, may be related to their low self-esteem and impairment of cognitive processes, which, as a result, contributes to their lower quality of life.

Adolescents diagnosed with behavioural and emotional disturbances who consume alcohol are distinguished by significantly higher levels of hostility, paranoid thinking, physical aggression, as well as a lower quality of life. It is possible that these minors intoxicate themselves with alcohol in order to regulate their emotions, as they need to reduce their frustration resulting from the unfulfilled need for safety in their dysfunctional families.

The more severe the psychopathological symptoms, the more severe the aggression. The current study provides no grounds to indicate that there is a causal relationship between them.

## Figures and Tables

**Table 1 ijms-24-07097-t001:** Comparative analysis of psychological characteristics of adolescents, depending on gender and age.

Parameter/Trait	Girls n = 82	Boys n = 41	*p* *	Aged 13–14 Years n = 30	Aged 15–18 Years n = 93	*p* *
	Mean ± SD			Mean ± SD		
SOM_SCL	1.1 ± 1.0	0.6 ± 0.6	0.0087	0.7 ± 0.8	1.1 ± 0.9	0.0072
OC_SCL	1.5 ± 1.1	1.1 ± 0.9	0.077	1.0 ± 1.1	1.5 ± 1.0	0.013
SENS_SCL	1.6 ± 1.0	1.2 ± 1.0	0.037	1.3 ± 1.2	1.5 ± 1.0	0.11
DEP_SCL	1.9 ± 1.1	1.4 ± 0.9	0.015	1.5 ± 1.1	1.9 ± 1.0	0.12
ANX_SCL	1.4 ± 1.1	1.0 ± 0.9	0.12	0.9 ± 1.0	1.4 ± 1.1	0.0069
ANG_SCL	1.5 ± 1.0	1.2 ± 1.0	0.089	1.3 ± 1.0	1.4 ± 1.0	0.60
PHO_SCL	1.1 ± 1.0	0.7 ± 0.8	0.056	0.7 ± 1.0	1.0 ± 1.0	0.10
PAR_SCL	1.5 ± 1.0	1.2 ± 1.1	0.15	1.2 ± 1.2	1.5 ± 1.0	0.12
PSY_SCL	1.1 ± 0.9	0.7 ± 0.7	0.070	0.8 ± 0.9	1.0 ± 0.9	0.12
A_BP	21.1 ± 5.9	20.7 ± 5.5	0.47	20.5 ± 6.2	21.1 ± 5.6	0.81
PA_BP	22.7 ± 8.6	25.8 ± 8.1	0.051	21.9 ± 7.8	24.3 ± 8.7	0.26
H_BP	25.5 ± 6.9	25.1 ± 6.7	0.83	24.9 ± 6.7	25.5 ± 6.9	0.71
VA_BP	15.3 ± 4.5	15.4 ± 3.8	0.89	14.4 ± 5.0	15.6 ± 4.0	0.24
SUM_BP	84.6 ± 20.9	87.0 ± 19.8	0.64	81.6 ± 21.3	86.6 ± 20.2	0.37

* Mann–Whitney U test, SD—standard deviation.

**Table 2 ijms-24-07097-t002:** Comparative analysis of psychological characteristics of adolescents, depending on attempted suicide/unattempted suicide.

Parameter/Trait	Without Suicide Attempts n = 78	With Suicide Attempts n = 45	*p* *
	Mean ± SD		
SOM_SCL	0.8 ± 0.9	1.2 ± 0.9	0.010
OC_SCL	1.2 ± 1.0	1.6 ± 1.1	0.051
SENS_SCL	1.4 ± 1.1	1.6 ± 1.0	0.13
DEP_SCL	1.6 ± 1.0	2.1 ± 1.1	0.013
ANX_SCL	1.1 ± 1.0	1.6 ± 1.1	0.0034
ANG_SCL	1.3 ± 0.9	1.6 ± 1.0	0.13
PHO_SCL	0.9 ± 1.0	1.1 ± 0.9	0.081
PAR_SCL	1.3 ± 1.1	1.6 ± 1.0	0.041
PSY_SCL	0.8 ± 0.8	1.2 ± 1.0	0.018
A_BP	20.2 ± 5.6	22.3 ± 5.8	0.054
PA_BP	23.9 ± 8.0	23.5 ± 9.4	0.71
H_BP	24.4 ± 7.0	27.1 ± 6.3	0.042
VA_BP	15.1 ± 4.0	15.8 ± 4.7	0.39
SUM_BP	83.5 ± 20.0	88.7 ± 21.1	0.22

* Mann–Whitney U test, SD—standard deviation.

**Table 3 ijms-24-07097-t003:** Comparative analysis of selected psychological traits of adolescents, depending on alcohol abuse in the family and the occurrence of mental illness in family members. o statistically.

Parameter/Trait	Alcohol Abuse in the Family NO n = 62	Alcohol Abuse in the Family YES n = 61	*p* *	Mental Illness in the Family NO n = 87	Mental Illness in the Family YES n = 36	*p* *
	Mean ± SD			Mean ± SD		
SOM_SCL	1.1 ± 1.0	0.8 ± 0.8	0.084	0.9 ± 0.8	1.1 ± 1.0	0.76
OC_SCL	1.6 ± 1.1	1.1 ± 1.0	0.015	1.4 ± 1.0	1.3 ± 1.2	0.20
SENS_SCL	1.7 ± 1.0	1.3 ± 1.0	0.026	1.5 ± 1.0	1.4 ± 1.1	0.53
DEP_SCL	2.0 ± 1.0	1.5 ± 1.0	0.017	1.9 ± 1.0	1.5 ± 1.2	0.092
ANX_SCL	1.4 ± 1.0	1.2 ± 1.1	0.13	1.3 ± 1.0	1.3 ± 1.2	0.59
ANG_SCL	1.4 ± 1.1	1.3 ± 0.9	0.79	1.4 ± 0.9	1.3 ± 1.1	0.19
PHO_SCL	1.1 ± 0.9	0.8 ± 1.0	0.015	0.9 ± 0.9	0.9 ± 1.2	0.32
PAR_SCL	1.5 ± 1.1	1.3 ± 1.0	0.33	1.4 ± 1.0	1.3 ± 1.1	0.34
PSY_SCL	1.1 ± 0.9	0.8 ± 0.9	0.034	0.9 ± 0.8	1.0 ± 1.1	0.63
A_BP	21.4 ± 6.2	20.6 ± 5.3	0.48	21.4 ± 5.4	20.0 ± 6.5	0.30
PA_BP	23.2 ± 8.7	24.2 ± 8.4	0.41	23.8 ± 8.2	23.5 ± 9.5	0.67
H_BP	25.2 ± 7.5	25.6 ± 6.2	0.73	25.5 ± 6.3	25.0 ± 8.1	0.87
VA_BP	15.5 ± 4.3	15.1 ± 4.3	0.55	15.2 ± 4.3	15.6 ± 4.3	0.55
SUM_BP	85.3 ± 22.6	85.5 ± 18.3	0.68	85.9 ± 18.7	84.1 ± 24.5	0.66

* Mann–Whitney U test, SD—standard deviation.

**Table 4 ijms-24-07097-t004:** Analysis of Spearman’s rank correlation between psychological traits in patients under study.

	A_BP	PA_BP	H_BP	VA_BP	SUM_BP
SOM_SCL	**0.47**	0.17	**0.38**	**0.27**	**0.38**
OC_SCL	**0.42**	0.10	**0.49**	**0.26**	**0.37**
SENS_SCL	**0.42**	0.16	**0.51**	**0.28**	**0.41**
DEP_SCL	**0.38**	0.07	**0.49**	**0.19**	**0.34**
ANX_SCL	**0.48**	0.15	**0.47**	**0.30**	**0.41**
ANG_SCL	**0.60**	**0.45**	**0.58**	**0.43**	**0.65**
PHO_SCL	**0.34**	0.05	**0.43**	**0.18**	**0.28**
PAR_SCL	**0.45**	**0.28**	**0.59**	**0.33**	**0.51**
PSY_SCL	**0.45**	0.16	**0.57**	**0.31**	**0.45**

## Data Availability

Not applicable.

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
