# Peer review of "Association between Psychopathological Symptoms and Aggression and Selected Biochemical Parameters in Adolescents with Behavioural and Emotional Disturbances"

_ijms, 2023, doi:10.3390/ijms24087097_

Round 1

Reviewer 1 Report

I have several comments.

1) Introduction

Line 75-84: Is this the content of the article or someone else's comment? It looks like someone pasted a comment, needs to be deleted.

2) Materials and Methods

Were the ages of the boys and girls different from each other? This should be reflected if there was no difference in age.

3) Results

Line 103-106: А comment on the difference between boys relative to girls in biochemical parameters (potassium, creatinine ALT, AST) is given and P-values are given, but it is necessary to give absolute value with units of measurement this parameters. 

4) Discussion: Please write a section about limitations of the study (sample size). 

Could the data be influenced by genetics? In particular, it is known that the rs6265 variant of the BDNF gene is often associated with risky behavior, subclinical depression (Haslacher H et al, 2015), etc.? What do the authors think about it? https://pubmed.ncbi.nlm.nih.gov/25998702/ 

Reviewer 2 Report

The control group is missing. Children already diagnosed with specific disorders are studied. There is no comparison of BDNF with a group of children who have no diagnosis or have been tested but are considered healthy.

Since the aim of the study is to assess the quality of life and psychosomatic symptoms, as well as cortisol and BDNF concentrations, there is no comparison with healthy children.

No indication of a specific problem with respect to scientific research on this topic.

In the present study, the research results are presented very accurately, for a wide group of respondents.

- not all results are presented in the table (the author informs about it).

Disproportions in the number of girls to boys. The same is true for the functional and dysfunctional families studied. Unequal disproportion in functional to dysfunctional families.
